# Exogenous NAD^+^ Stimulates MUC2 Expression in LS 174T Goblet Cells via the PLC-Delta/PTGES/PKC-Delta/ERK/CREB Signaling Pathway

**DOI:** 10.3390/biom10040580

**Published:** 2020-04-09

**Authors:** Seongho Ma, Jiah Yeom, Young-Hee Lim

**Affiliations:** 1Department of Integrated Biomedical and Life Sciences, Graduate School, Korea University, Seoul 02841, Korea; aktjdgh8@naver.com (S.M.); intro56@naver.com (J.Y.); 2Department of Public Health Science (Brain Korea 21 PLUS program), Graduate School, Korea University, Seoul 02841, Korea; 3Department of Laboratory Medicine, Korea University Guro Hospital, Seoul 08308, Korea

**Keywords:** NAD^+^, MUC2, goblet cell, cyclic AMP response element-binding protein, arachidonic acid, phospholipase C

## Abstract

Background: MUC2, a major component of the mucus layer in the intestine, is associated with antimicrobial activity and gut immune system function. Currently, mucin is mainly known for its critical function in defense against toxic molecules and pathogens. In this study, we investigated the stimulatory effects of exogenous nicotinamide adenine dinucleotide (NAD^+^) on the expression of MUC2 in LS 174T goblet cells. Methods: Genes related to MUC2 synthesis were measured by quantitative real-time PCR (qPCR). To analyze the gene expression profiles of NAD^+^-treated LS 174T goblet cells, RNA sequencing was performed. MUC2 expression in the cells and secreted MUC2 were measured by immunocytochemistry (ICC) and ELISA, respectively. Results: NAD^+^ significantly stimulated MUC2 expression at mRNA and protein levels and increased the secretion of MUC2. Through RNA sequencing, we found that the expression of genes involved in arachidonic acid metabolism increased in NAD^+^-treated cells compared with the negative control cells. NAD^+^ treatment increased phospholipase C (PLC)-δ and prostaglandin E synthase (PTGES) expression, which was inhibited by the appropriate inhibitors. Among the protein kinase C (PKC) isozymes, PKC-δ was involved in the increase in MUC2 expression. In addition, extracellular signal-regulated kinase (ERK)1/2 and cyclic AMP (cAMP) response element-binding protein (CREB) transcript levels were higher in NAD^+^-treated cells than in the negative control cells, and the enhanced levels of phosphorylated CREB augmented MUC2 expression. Conclusions: Exogenous NAD^+^ increases MUC2 expression by stimulating the PLC-δ/PTGES/PKC-δ/ERK/CREB signaling pathway.

## 1. Introduction

The mucus layer in the intestine is mainly composed of mucin glycoproteins that are secreted from intestinal goblet cells. The mucus layer acts as an intestinal barrier and plays a critical role in preventing toxic molecules and pathogens from penetrating into the intestinal mucosae, thus preventing intestinal inflammation [1]. For this reason, the intestinal mucus layer is the first line of immune defense protecting a host from pathogens. Destruction of the mucus layer causes severe intractable inflammatory bowel diseases (IBDs), including Crohn’s disease and ulcerative colitis [2]. In addition to damage to the mucus layer, decreases in mucin secretion by intestinal goblet cells can contribute to chronic inflammation in the intestine [3]. Therefore, conservation of an intact mucus layer that properly regulates immune responses is a main factor that is necessary for the maintenance of intestinal homeostasis and the prevention of intestinal inflammatory diseases.

MUC2 gene expression is regulated by cyclic AMP (cAMP) response element-binding protein (CREB) in retinoic acid-treated normal human tracheobronchial epithelial (NHTBE) cells [4]. CREB is a transcription factor regulating the expression of genes involved in a variety of cellular processes, such as cell growth and survival, differentiation, and mucin production [5]. Upon cAMP-induced phosphorylation through intracellular signaling cascades, CREB is activated and stimulates gene transcription by binding to the promoters of target genes. CREB phosphorylation through the mitogen-activated protein kinase (MAPK) signaling pathway is associated with various human diseases [5]. For example, angiotensin II induces cardiac fibrosis by enhancing periostin expression via the p38 MAPK–CREB pathway [6]. The protein kinase C (PKC)/p38 MAPK/CREB pathway is involved in membrane wound repair [7]. An E3 ubiquitin ligase (CUL4A) promotes the proliferation of cancer cells through the phosphorylation-mediated activation of ERK1/2 followed by CREB [8]. Paeoniflorin, derived from the herbal plant *Paeonia lactiflora*, exerts neuroprotective effects via the ERK–CREB signaling pathway in rats with hippocampal damage [9].

Nicotinamide adenine dinucleotide (NAD^+^) belongs to the family of natural purine compounds. NAD^+^ is a coenzyme that transfers electrons in the electron transport chain (ETC) and plays an essential role in cellular metabolism. NAD^+^ can be biosynthesized using amino acids such as aspartate and tryptophan and can be recycled through salvage pathways [10]. This coenzyme plays major roles in many redox reactions [11] and has also been revealed to participate in non-redox reactions in some cases. NAD^+^, a substrate for poly(ADP-ribose) polymerases (PARPs), is catabolized to transfer ADP-ribose moieties to other proteins in response to DNA damage or during transcriptional regulation and energy metabolism [12]. NAD^+^ binds to sirtuins called histone deacetylases (HDACs), whose main function is to reverse the acetyl modification of histones. A second messenger called cyclic ADP-ribose (cADPR), a form of NAD^+^ converted by ADP-ribosyl cyclase [13], is a messenger in calcium signaling. Recently, in addition to its various intracellular roles, extracellular NAD^+^ has been found to mediate intracellular signaling pathways by acting as an agonist of purine receptors [14,15,16]. For example, ADP-ribose produced from NAD^+^ is an agonist of the P2 purinergic receptor P2Y1 [17]. The family of P2 purinergic receptors is mainly coupled with G proteins that activate phospholipase C (PLC) [18,19].

In our previous study, we found that oxyresveratrol, an antioxidant stilbenoid, stimulated MUC2 synthesis and increased NAD^+^ levels by activating enzymes involved in the salvage pathways of NAD^+^ biosynthesis in LS 174T goblet cells, revealing that NAD^+^ plays a novel role in the formation of an intact intestinal mucus layer by stimulating mucin production [20]. Based on these data, we hypothesized that NAD^+^ might be involved in MUC2 synthesis in the intestine. Thus, in this study, to elucidate the mechanism of MUC2 production by NAD^+^, we investigated the signaling pathway of MUC2 production upon the administration of exogenous NAD^+^, and we found that NAD^+^ increased MUC2 expression by stimulating the PLC-δ/PTGES/PKC-δ/ERK/CREB signaling pathway.

## 2. Materials and Methods

### 2.1. Materials

Roswell Park Memorial Institute (RPMI) 1640 medium, fetal bovine serum (FBS), and penicillin/streptomycin for culture of LS 174T cells were obtained from HyClone (Logan, UT, USA). 3-(4,5-Dimethylthiazol-2-yl)-2,5-diphenyltetrazolium bromide (MTT) was obtained from Amresco (Solon, OH, USA). All the chemical inhibitors used in this study were from Selleckchem (Houston, TX, USA), except for MF63 (Cayman Chemical, Ann Arbor, MI, USA) and AS-65111 (Anaspec, Fremont, CA, USA). NAD^+^, dimethyl sulfoxide (DMSO), and 4′,6-diamidino-2-phenylindole (DAPI) were obtained from Sigma (St. Louis, MO, USA).

### 2.2. Cell Culture

The LS 174T goblet cell line was obtained from the Korean Cell Line Bank (Seoul, Korea) and cultured in a 90-mm culture dish with 10 mL RPMI 1640 medium supplemented with 10% FBS, 100 units/mL penicillin, and 100 μg/mL streptomycin at 37 °C in an atmosphere of 5% CO_2_ and 95% air. The cells were seeded at the concentration of 2.5 × 10^5^ cells/mL. The culture medium was replaced with fresh medium every two days. Chemical or peptide inhibitors were administered alone or in combination with 200 µM NAD^+^ for 48 h. The inhibitors U73122, MF63, AS-65111, U0126, SB203580, SP600125, and KG-501 were used to block PLC-δ, prostaglandin E synthase (PTGES), PKC-δ, mitogen-activated protein kinase (MEK)/extracellular signal-regulated kinase (ERK), p38, Jun-N-terminal kinase (JNK,) and CREB at final concentrations of 10 µM, 500 nM, 40 µM, 100 µM, 5 µM, 10 μM, and 10 µM, respectively, according to the manufacturers’ guidelines.

### 2.3. Assessment of Cytotoxicity of Exogenous NAD^+^ to LS 174T Cells

LS 174T cells were seeded in 96-well plates and treated with NAD^+^ at final concentrations of 50, 100, and 200 µM in RPMI medium for 48 h. After the medium was removed, the cells were incubated in MTT diluted with the medium for 1 h at 37 °C. The produced formazan crystals were solubilized with DMSO for 1 h at room temperature. The absorbance of each sample was measured at 540 nm with a SpectraMax 340 microplate reader (Molecular Devices Corp., Sunnyvale, CA, USA). The relative cell viability (%) was calculated with the following equation: cell viability (%) = [OD (experiment group)/OD (control group)] × 100.

### 2.4. Quantitative Real-Time Polymerase Chain Reaction (qPCR)

The total RNA from each sample was extracted with RiboEx (Geneall, Seoul, Korea) according to the manufacturer’s instructions, quantified with a NanoDrop ND-1000 spectrophotometer (Thermo Scientific, Wilmington, DE, USA), and converted to cDNA using a RevertAid First Strand cDNA Synthesis kit (Thermo Fisher Scientific, Waltham, MA, USA). qPCR was performed with a Kapa SYBR Fast qPCR Master Mix kit (Kapa Biosystems, Woburn, MA, USA) and a StepOnePlus Real-Time PCR System (Applied Biosystems, Foster City, CA, USA). The PCR primer sequences are shown in Table 1. Each qPCR sample was preheated at 95 °C for 10 min and then subjected to 40 cycles of 95 °C for 15 s, 60 °C for 15 s, and 72 °C for 30 s. *β-Actin* was used as an internal control gene. The data were analyzed based on the 2^-ΔΔCt^ method (the *β-actin* control was set to 1) [21].

### 2.5. Immunocytochemistry (ICC)

The cells were fixed with 4% paraformaldehyde at 4 °C overnight. A permeabilization step with 0.1% Triton X-100 was performed for 10 min followed by blocking with 10% normal donkey serum (GTX73205, Genetex, Irvine, CA, USA). The cells were incubated with a primary antibody against MUC2 (1:1000 dilution; GTX100664, Genetex) at 4 °C overnight. Goat anti-rabbit immunoglobulin G (IgG) (H + L), DyLight 488 (1:1000 dilution; 35552, Thermo Scientific, Wilmington, DE, USA), was used as the secondary antibody for green fluorescence, and DAPI (1:10,000 dilution) was used to counterstain the nuclei for 5 min at room temperature. After washing samples with phosphate-buffered saline (PBS) three times for 5 min each, the slides were mounted using VECTASHIELD^®^ (Vector Laboratories, Burlingame, CA, USA). Images were captured using a Nikon C1 plus confocal laser scanning microscope (Nikon, Tokyo, Japan), and the intensity of green fluorescence indicating MUC2 protein was quantified by Olympus fluoview FV1000 ver.2.1b (Olympus, Tokyo, Japan) followed by normalizing the detected area and expressed as (%) compared to the negative control (100%).

### 2.6. ELISA

To measure secreted MUC2, cultured LS 174 cells treated with NAD^+^ and/or MF63 for 48 h were centrifuged at 1000× *g* at 4 °C for 20 min, and the supernatants were stored at −80 °C until use. The concentration of secreted MUC2 protein in the supernatants was measured following the instructions of a Human MUC2 ELISA Kit obtained from Elabscience (E-EL-H0632, Houston, TX, USA).

### 2.7. RNA Sequencing

The total RNA from the negative control and 200 μM NAD^+^-treated cells was extracted with RiboEx (GeneAll, Seoul, Korea) according to the manufacturer’s instructions, and RNA purity and integrity were measured with a NanoDrop ND-1000 spectrophotometer and an Agilent 2100 Bioanalyzer (Agilent Technologies, Santa Clara, CA, USA), respectively. cDNA libraries were constructed using a QuantSeq 3′ mRNA-Seq Library Prep Kit (Lexogen, Greenland, NH, USA) according to the manufacturer’s instructions, and 75 bp single-end sequencing was performed on an Illumina NextSeq 500 platform (San Diego, CA, USA). The RNA-seq data have been deposited in NCBI’s Gene Expression Omnibus (GEO) and are accessible through GEO Series accession numbers GSM4154599-GSM4154604 (accession code: GSE140116). All data were obtained after quantile normalization between samples, and differentially expressed genes (DEGs) were identified with Excel-based Differentially Expressed Gene Analysis (ExDEGA) version 2.0.0 provided by eBiogen (Seoul, Korea). The functional annotation of 648 genes filtered with a criterion of at least a 2.0-fold change with a *p*-value < 0.05 was performed using the Database for Annotation, Visualization and Integrated Discovery (DAVID) (http://david.ncifcrf.gov/summary.jsp). We identified some pathways that were predicted to be involved; then, to more specifically observe the network among the DEGs, the 648 genes were used as input for Search Tool for the Retrieval of Interacting Genes/Proteins (STRING) analysis (http://string-db.org).

### 2.8. Western Blot Analysis

LS 174T cells were seeded and treated with 200 μM NAD^+^ in 90 mm dishes for 48 h. After being briefly washed with cold PBS, the cells were harvested in cold PBS by scraping on ice and centrifuged at 3000× *g* for 3 min. Protein was extracted by lysing the cells with a Pro-Prep Kit (Intron Biotechnology, Seongnam, Gyeonggi-do, Korea). The protein concentration was measured by Bradford assay, and equal amounts of protein (20 µg) from the different samples were separated by 10% sodium dodecyl sulfate-polyacrylamide gel electrophoresis (SDS-PAGE). Subsequently, the separated proteins were transferred to a polyvinylidene difluoride (PVDF) membrane (Millipore, Bedford, MA, USA) with a Trans-Blot semi-dry transfer cell (Bio-Rad, Hercules, CA, USA). The membrane was blocked with 5% skim milk (Neogen, Lansing, MI, USA) for total CREB analysis or with 5% bovine serum albumin (BSA, Sigma) for phosphorylated CREB analysis in Tris-buffered saline (TBS) containing 0.1% Tween 20 (TBS-T) and incubated with anti-CREB (1:1000 dilution; 48H2, Cell Signalling Technology, Danvers, MA, USA) and anti-phosphor-CREB (1:1000 dilution; 87G3, Cell Signalling Technology) primary antibodies at 4 °C overnight. The membrane was washed three times with TBS-T and reacted with the secondary antibody at room temperature for 1 h. Horseradish peroxidase-conjugated goat anti-rabbit IgG (H + L) (1:1000 dilution; NCI1460KR, Thermo Scientific) was used as the secondary antibody. The membrane was washed three times with TBS-T, and the protein bands were detected and analyzed after incubating the membrane with SuperSignal West Femto Maximum Sensitivity Substrate (34095, Thermo Scientific) using a FluoroChem E imaging system (ProteinSimple, San Jose, CA, USA). The densities of the bands were quantified using ImageJ version 1.52a (National Institutes of Health). The blots in the figures are representative of three independent experiments.

### 2.9. Statistical Analysis

All statistical analyses were performed using the Statistical Package for the Social Sciences (SPSS, Chicago, IL, USA) version 24.0. The data are expressed as the mean ± standard deviation (SD) of three independent experiments performed in triplicate. The significance of the differences between samples was calculated by Student’s t-test or one-way analysis of variance (ANOVA) followed by Tukey’s HSD test. A *p*-value of < 0.05 was considered significant.

## 3. Results

### 3.1. Cytotoxicity of Exogenous NAD^+^ to LS 174T Goblet Cells

To study the stimulatory effect of NAD^+^ on MUC2 expression in LS 174T cells, we measured the cytotoxic effect of exogenous NAD^+^ on LS 174T cells. The relative viability of the cells treated with NAD^+^ (12.5‒200 µM) was not significantly different from that of the negative control cells (Figure 1A). Therefore, exogenous NAD^+^ was not cytotoxic to LS 174T cells within the concentration range used in this study.

### 3.2. Time Course of MUC2 Induction by Exogenous NAD^+^

To determine the maximal time point of *MUC2* induction, the expression level of *MUC2* was measured by culture time by qPCR. Cells treated with 200 µM NAD^+^ showed higher expression rates compared with the negative control at each time point (Figure 1B). Especially, cells treated with 200 µM NAD^+^ showed the highest expression level at 48 h. Therefore, cells were cultured for 48 h in subsequent experiments in this study.

### 3.3. Effect of Exogenous NAD^+^ on the Expression of MUC2 in LS 174T Cells

The stimulatory effect of exogenous NAD^+^ on MUC2 expression in LS 174T cells was measured by quantitative real-time polymerase chain reaction (qPCR) (at the mRNA level) and by immunocytochemistry (ICC) (at the protein level). The expression levels of *MUC2* exhibited changes of 0.965-, 1.587-, and 2.305-fold in cells treated with 50 µM, 100 µM, and 200 µM NAD^+^, respectively, compared with the negative control cells (Figure 2A). MUC2 levels were also higher in NAD^+^-treated cells than in the negative control cells (Figure 2B). The quantified fluorescence intensity in NAD^+^-treated cells increased in a dose-dependent manner; the relative fluorescence intensities in 50 µM, 100 µM, and 200 µM NAD^+^-treated cells were 112.973 ± 16.903%, 143.600 ± 1.424%, and 190.683 ± 4.079%, respectively, of the intensities in the negative control cells (100%) (Figure 2C). The results showed that exogenous NAD^+^ increased MUC2 expression at the mRNA and protein levels.

### 3.4. Gene Expression Analysis of NAD^+^-Treated LS 174T Cells

To estimate the pathway involved in the stimulation of *MUC2* expression by exogenous NAD^+^ in LS 174T cells, RNA sequencing of the negative control and 200 µM NAD^+^-treated cells was performed in three independent experiments, and 25,737 genes were identified (Appendix A). RNA sequencing data were deposited and are available through the NCBI GEO database (accession code: GSE140116). For functional annotation, Kyoto Encyclopedia of Genes and Genomes (KEGG) enrichment analysis was performed with the Database for Annotation, Visualization and Integrated Discovery (DAVID). Arachidonic acid metabolism (−log(*P*-value) = 7.500), steroid hormone biosynthesis (−log(*P*-value) = 3.997), and glycine, serine, and threonine metabolism (−log(*P*-value) = 3.380) were the most significantly enriched Gene Ontology (GO) terms of the differentially expressed genes (DEGs) in NAD^+^-treated LS 174T cells compared with the negative control cells (Figure 3A). We further analyzed mRNA expression fold changes in the cells treated with 200 μM NAD^+^ compared with the negative control cells. Interestingly, the levels of several small nucleolar RNAs (snoRNAs) were highly increased in the cells treated with 200 μM NAD^+^ compared with the negative control cells. Among the DEGs, 648 genes changed by ≥ 2-fold or ≤ 0.5-fold with statistical significance (*p* < 0.05) upon NAD^+^ treatment (Appendix A). The volcano plot shows the 451 upregulated genes (≥ 2-fold) (red) and the 197 downregulated genes (≤ 0.5-fold) (green) in NAD^+^-treated cells compared with the negative control cells (Figure 3B), and the heatmap further indicates the upregulated (red) and downregulated (blue) genes in the NAD^+^-treated cells compared with the negative control cells and the genes involved in arachidonic acid metabolism (Figure 3C). Among 13 genes involved in arachidonic acid metabolism, 12 genes (except *PTGS2* (*COX-2*)) were significantly upregulated by exogenous NAD^+^ (Table 2). The arachidonic acid metabolism network was obtained through STRING analysis (Figure 3D).

### 3.5. The Expression of Phospholipase A2 (PLA_2_) is Increased in NAD^+^-Treated Cells

Cytosolic PLA_2_ (cPLA_2_) converts membrane phospholipids into arachidonic acid [22]. To confirm the effect of NAD^+^ treatment on *PLA_2_* expression, expression levels of *PLA_2_* were measured by qPCR. The expression levels of *PLA_2_* were significantly increased in a dose-dependent manner in NAD^+^-treated cells compared with the negative control cells (Figure 4). The expression levels of *PLA_2_* in cells treated with 50 µM, 100 µM, and 200 µM NAD^+^ were 1.191-, 1.329-, and 1.677-fold higher, respectively, than those in the negative control cells.

### 3.6. Phospholipase C (PLC) is Involved in NAD^+^-Induced Increases in MUC2 Expression

Extracellular NAD^+^ induces signaling cascades by activating PLC via P2 receptors/Gq proteins [18,19]. To examine whether exogenous NAD^+^ induces PLC activation, we measured the expression levels of *PLC-δ3*, *PLC-γ2*, and *PLC-η2* in NAD^+^-treated cells by qPCR. *PLC-δ3*, *PLC-γ2*, and *PLC-η2* were selected based on the RNA sequencing data (Appendix A), and their expression levels were ≥ 1.5-fold higher (*p* < 0.05) in NAD^+^-treated cells compared with the negative control cells. Among *PLC-δ3*, *PLC-γ2*, and *PLC-η2*, *PLC-δ3* exhibited significant increases in expression in the NAD^+^-treated cells compared with the negative control cells (Figure 5A). The expression levels of *PLC-δ3* in cells treated with 50 µM, 100 µM, and 200 µM NAD^+^ were 0.936-, 2.197-, and 4.089-fold higher, respectively, than those in the negative control cells. To confirm the effect of PLC on MUC2 expression mediated by NAD^+^, we administered U73122, a potent PLC inhibitor, along with NAD^+^ and measured the expression levels of *MUC2*. The expression levels of *MUC2* exhibited changes of 0.995-, 1.551-, and 0.874-fold in cells treated with 10 μM U73122 alone, 200 µM NAD^+^, and U73122 and 200 µM NAD^+^, respectively, compared with the negative control cells (Figure 5B). U73122 alone did not affect *MUC2* expression levels; however, the co-administration of U73122 with 200 µM NAD^+^ significantly suppressed *MUC2* expression. In ICC analysis, co-administration of U73122 and 200 µM NAD^+^ decreased MUC2 levels (Figure 5C). In cells treated only with the secondary antibody (and not with the primary anti-MUC2 antibody), the fluorescence intensities did not increase, indicating that the fluorescence intensity was not affected by the secondary antibodies. Compared with the levels in the negative control cells (100%), the expression levels of MUC2 protein were 93.630 ± 2.287%, 167.568 ± 8.929%, and 91.565 ± 5.936% in cells treated with U73122 alone, 200 µM NAD^+^, and U73122 and 200 µM NAD^+^, respectively (Figure 5D). The expression levels of MUC2 protein in the different samples were consistent with the gene expression results. The results suggested that PLC-δ3 might be involved in the stimulation of MUC2 expression by exogenous NAD^+^.

### 3.7. Microsomal Prostaglandin E Synthase (PTGES) is Involved in the Stimulation of MUC2 Expression by Exogenous NAD^+^

Mucus secretion is stimulated by gastrointestinal hormones such as prostaglandin E2 (PGE_2_) [23,24], and PGE_2_ is produced from arachidonic acid by PTGES. The expression levels of *PTGES* in 100 and 200 µM NAD^+^-treated cells were significantly increased (50 µM NAD^+^, 0.881-fold; 100 µM NAD^+^, 1.690-fold; 200 µM NAD^+^, 3.054-fold, compared with the levels in the negative control cells) (Figure 6A). To confirm these results, we used MF63, a potent selective inhibitor of human PTGES, to block the activity of PTGES in this pathway. The expression levels of *MUC2* were altered by 0.467-, 1.805-, and 0.716-fold in cells treated with 500 nM MF63 alone, 200 µM NAD^+^, and 200 µM NAD^+^ and MF63, respectively, compared with the negative control cells (Figure 6B). To investigate whether the inhibition of PTGES by MF63 affected MUC2 expression, we measured the protein expression levels of secreted MUC2 in the cell culture supernatant by ELISA rather than by ICC, because MF63 emits green fluorescence in the absence of primary and secondary antibodies due to its structural character. The expression levels of MUC2 were 12.027 ± 1.296 ng/mL in the negative control cells and 13.840 ± 0.404 ng/mL, 14.540 ± 0.256 ng/mL, 15.082 ± 0.159 ng/mL, 9.260 ± 0.100 ng/mL, and 10.298 ± 0.059 ng/mL in the cells treated with 50 µM NAD^+^, 100 µM NAD^+^, 200 µM NAD^+^, MF63 alone, and 200 µM NAD^+^ and MF63, respectively (Figure 6C). PGE_2_ stimulates mucin secretion, and PTGES is a key enzyme for PGE_2_ synthesis, which suggests that the inhibition of PTGES might decrease MUC2 production. In this study, although the *MUC2* gene expression was not significantly decreased, MF63 alone significantly decreased the expression level of MUC2 protein in LS 174T, which may be due to the inhibition of PTGES, which resulted in the decrease of MUC2 expression. The results showed a small but significant increase of secreted MUC2 in a dose-dependent manner with NAD^+^ and suggested that PTGES might contribute to the stimulation of MUC2 expression by exogenous NAD^+^.

### 3.8. PKC-δ in the MUC2 Expression Pathway Receives a Signal upon NAD^+^ Administration

In general, PGE_2_ is known for binding to the PGE2 receptor, which induces the activation of PKC [25]. To observe the effect of NAD^+^ on the mRNA expression of PKC in NAD^+^-treated cells, we performed qPCR. Among the PKC isotypes, PKC-*δ* showed the greatest dose-dependent increases in expression in our preliminary experiments (Figure 7A). The expression levels of *PKC-δ* in the cells treated with 50 µM, 100 µM, and 200 µM NAD^+^ were 1.125-, 1.549-, and 2.578-fold higher, respectively, than those in the negative control cells. For the specific inhibition of PKC-δ, the peptide AS-65111 was used. AS-65111 (40 μM) alone did not block MUC2 at either the gene level (Figure 7B) or the protein level (Figure 7C,D); however, it successfully attenuated the increases in MUC2 expression in 200 µM NAD^+^-treated cells at the gene and protein levels. The expression levels of *MUC2* were 1.095-, 1.573-, and 1.072-fold higher in the cells treated with AS-65111 alone, 200 µM NAD^+^, and 200 µM NAD^+^ and AS-65111, respectively, than in the negative control cells (Figure 7B). The expression levels of MUC2 were increased by treatment with 200 µM NAD^+^ but decreased by co-treatment with 200 µM NAD^+^ and AS-65111 (Figure 7C,D). The expression levels of MUC2 were 106.637 ± 1.165%, 239.137 ± 7.217%, and 147.589 ± 0.460% of the negative control levels in the cells treated with AS-65111 alone, 200 µM NAD^+^, and 200 µM NAD^+^ and AS-65111, respectively. The results suggested that PKC-δ might contribute to MUC2 expression through PLC stimulation by exogenous NAD^+^.

### 3.9. PKC-δ Transfers a Signal to ERK1/2 to Stimulate MUC2 Expression upon NAD^+^ Administration

To identify the signaling protein downstream from PKC-δ, the expression levels of genes encoding typical MAPK proteins (p38, ERK, and JNK) were measured. Among MAPK proteins, although expression levels were not large, *ERK1/2* exhibited a significant increase in expression in the cells treated with 100 µM and 200 µM NAD^+^, respectively, compared with the negative control cells (Figure 8A). The results showed that expression levels of JNK and p38 did not increase significantly by exogenous NAD^+^ supplementation. To confirm these results, the inhibitors U0126, SB203580, and SP600125 were used to block MEK/ERK, p38, and JNK, respectively. U0126 inhibited *MUC2* expression in 200 µM NAD^+^-treated cells (Figure 8B). The relative mRNA expression levels were 1.127 ± 0.350, 1.831 ± 0.113, and 0.815 ± 0.233 in the cells treated with U0126 alone, 200 µM NAD^+^, and 200 µM NAD^+^ and U0126, respectively, compared with the negative control cells (1.000). However, SB203580 and SP600125 did not inhibit *MUC2* expression in 200 µM NAD^+^-treated cells (Figure 8C,D). ICC showed that MUC2 expression increased in the cells treated with 200 µM NAD^+^; however, it decreased upon co-treatment with 200 µM NAD^+^ and U0126 (Figure 8E). The quantified relative protein expression levels of ICC were 102.795 ± 7.339%, 174.972 ± 28.216%, and 134.465 ± 14.138% of the negative control levels (100%) in the cells treated with U0126 alone, 200 µM NAD^+^, and 200 µM NAD^+^ and U0126, respectively (Figure 8F). These results suggested that among MAPK proteins, ERK might be involved in the pathway of MUC2 synthesis mediated by PKC-δ upon exogenous NAD^+^ administration.

### 3.10. CREB Receives a Signal from ERK and Becomes Phosphorylated

CREB is known as a transcription factor involved in prostaglandin-induced mucin synthesis [4,26]. To confirm the activation of the transcription factor CREB, the expression levels and phosphorylation of CREB were assayed. Indeed, the expression levels of *CREB* were significantly increased by 1.157-, 1.492-, and 2.787-fold in the cells treated with 50 µM, 100 µM, and 200 µM NAD^+^, respectively, compared with the negative control cells (Figure 9A). NAD^+^ also significantly increased the phosphorylation of CREB in a dose-dependent manner (Figure 9B,C). The phosphor-CREB/CREB ratios were 1.054 ± 0.133, 1.248 ± 0.056, and 2.127 ± 0.348 in the cells treated with 50 µM, 100 µM, and 200 µM NAD^+^, respectively, compared with the negative control cells (1.000). A CREB inhibitor, KG-501 (10 μM), was used to block MUC2 expression at the gene (Figure 9D) and protein levels (Figure 9E,F); the relative mRNA expression levels were 0.67050 ± 0.033, 1.876 ± 0.035, and 1.110 ± 0.040 in the cells treated with KG-501 alone, NAD^+^ 200 µM, and NAD^+^ 200 µM and KG-501, respectively, compared with the negative control cells. The quantified relative protein expression levels revealed by ICC were 107.927 ± 2.010%, 232.965 ± 24.341%, and 147.086 ± 6.492% of the negative control levels (100.000 ± 4.726%) in the cells treated with KG-501 alone, 200 µM NAD^+^, and 200 µM NAD^+^ and KG-501, respectively. Co-treatment with 200 µM NAD^+^ and KG-501 significantly decreased MUC2 expression at the gene and protein levels compared with 200 µM NAD^+^-only treatment. The results suggested that CREB might be a transcription factor in the pathway of MUC2 synthesis mediated by PKC upon exogenous NAD^+^ administration.

## 4. Discussion

NAD^+^ is commonly known for its role as a coenzyme for oxidoreductase in the context of redox metabolism [27]. However, NAD^+^ is also associated with ADP-ribose transfer reactions; in particular, in cell signaling, NAD^+^ is a precursor of cyclic ADP-ribose involved in the second messenger system [28,29]. Although NAD^+^ has been studied in a variety of fields in biology, it has not been reported to induce MUC2 expression. In this study, we found that exogenous NAD^+^ stimulates MUC2 expression in intestinal goblet cells, revealing a novel regulatory role of NAD^+^ in the synthesis of MUC2 in the intestine.

Extracellular NAD^+^ is an agonist of P2Y purinergic receptors [14,30]; these G protein-coupled receptors are linked to the activation of PLC followed by generation of the second messengers inositol 1,4,5-triphosphate (IP_3_) and diacylglycerol (DAG) [31]. NAD^+^/P2Y receptors transfer signals through PLC [19,32]. Purinergic receptors are required for extracellular NAD^+^ and evoke biological responses, which suggests that extracellular NAD^+^ may be sensed by cell-surface receptors. However, extracellular NAD^+^-binding receptors still remain unidentified. Through RNA sequencing, we found that the G protein-coupled receptor 87 (GPR87) and GPR34 genes, which are involved in the G protein-coupled purinergic receptor signaling pathway, were significantly upregulated by 9.807- and 3.360-fold, respectively, in NAD^+^-treated cells compared with control cells (Appendix A). GPR87 and GPR34 are P2Y-related receptor family members [33,34,35]. GPR87 is involved in various ligand-mediated signaling pathways, such as the MAPK pathway and PLC/PKC pathways activated by GPR87 [36]. The overexpression of GPR34 in lymphoma results in the phosphorylation of ERK, PKC, and CREB and increased cell proliferation [37].

Arachidonic acid is a polyunsaturated fatty acid present in the phospholipids of cellular membranes that is released by phospholipases. Arachidonic acid acts as a lipid secondary messenger in cellular signaling pathways involved in the regulation of PLC-γ and PLC-δ. Arachidonic acid strongly stimulates PKC-ε and PKC-δ in adult cardiac myocytes [38]. In this study, RNA sequencing showed that genes involved in arachidonic acid metabolism were upregulated in the cells treated with 200 μM NAD^+^. Arachidonic acid may be released from membrane phospholipids by PLC-δ and subsequently activate the signaling pathway that stimulates CREB, which might stimulate MUC2 expression. Similarly, CREB mediates MUC5AC overexpression induced by prostaglandin F_2α_ in airway epithelial cells through the PKC/ERK/p90 ribosomal protein kinase/CREB signaling cascade [26]. Quercetin increases MUC2 and MUC5AC gene expression and secretion via the PKCα/ERK1–2 pathway in LS 174T and Caco-2 cells [39]. Although we should measure the release of arachidonic acid upon NAD^+^ treatment in the further study, in this study, exogenous NAD^+^ showed the stimulating effect on MUC2 expression via involvement of PLC/PKC/MAPK pathway and CREB.

A secondary bile acid, deoxycholic acid, induces the transcription of *MUC2* in colon cancer cell line HM3, which is inhibited by the JNK-mediated pathway. Moreover, *MUC2* promoter activity is upregulated by inhibiting JNK with a JNK inhibitor, SP600125 [40]. In this study, SP600125 solely upregulated the transcriptional level of *MUC2*, which is consistent with those results. The promoters of *MUC2*, *MUC5AC*, and *MUC5B* contain putative CRE sites, and CREB knockdown decreases expression levels of *MUC2*, *MUC5AC*, and *MUC5B*, suggesting that CREB might be a potent transcriptional regulator of the mucin genes [4], which might be explained that a CREB inhibitor KG-501 alone decreased the expression level of *MUC2* compared with the negative control in this study.

In the functional annotation analysis, the arachidonic acid metabolism pathway was the most significant pathway. Arachidonic acid is converted to prostaglandins by cyclooxygenases (COX-1 and COX-2), and PGE_2_ is then produced by PTGES. COX-1 is constitutively expressed and produces prostaglandins that show homeostatic functions, such as gastric mucosal protection functions; in contrast, COX-2 shows inducible expression and produces prostaglandins that induce inflammation [41]. Arachidonic acid enables the production of PGE_2_, which mediates mucus secretion that contributes to ulcer healing in the gastrointestinal tract [42]. Through RNA sequencing, we found that among the genes involved in arachidonic acid metabolism, COX-2 was significantly downregulated, indicating that inflammation was not induced by exogenous NAD^+^.

Exogenous NAD^+^ triggered the activation of PLC-δ_3_, and the inhibition of PLC-δ activity decreased MUC2 expression. Activation of the phosphoinositide signaling cascade is involved in the activation of phospholipases. In this pathway, IP_3_ and DAG are released from the cell membrane by PLC. Generally, IP_3_ is known to diffuse and bind to calcium channels located in the endoplasmic reticulum (ER), while DAG is processed to form eicosanoids such as arachidonic acid by PLA_2_. PLA_2_, which is involved in arachidonic acid metabolism, was upregulated in NAD^+^-treated cells (Table 2). The expression level of *PTGES*, which converts arachidonic acid into PGE_2_, also increased upon treatment with NAD^+^. Increased levels of calcium ions and PGE_2_ may activate PKC-δ. Although the tyrosine phosphorylation of PKC-δ by thrombin is Ca^2+^-dependent [43], it is widely known that the activation of PKC-δ is Ca^2+^-independent, requiring only DAG [44,45,46]. In this study, NAD^+^ may have activated PKC-δ through signals transferred from PGE_2_ derived from DAG.

RNA sequencing revealed that several snoRNAs, which are noncoding RNAs, were upregulated by approximately 40- to 10-fold in the cells treated with 200 μM NAD^+^ compared with the negative control cells. These molecules were involved in the processing and modification of ribosomal RNAs. Further studies are needed to explain the large increases in the levels of these snoRNAs.

## 5. Conclusions

Exogenous NAD^+^ induces the activation of PLC-δ/PTGES/PKC-δ/ERK1/2 signaling to phosphorylate the transcription factor CREB, which ultimately enhances MUC2 expression in intestinal goblet cells. Therefore, NAD^+^ has the potential to improve intestinal mucosal barrier function by stimulating the expression of MUC2. Thus, our findings reveal a novel effect of NAD^+^ associated with non-redox reactions.

## Figures and Tables

**Figure 1 biomolecules-10-00580-f001:**
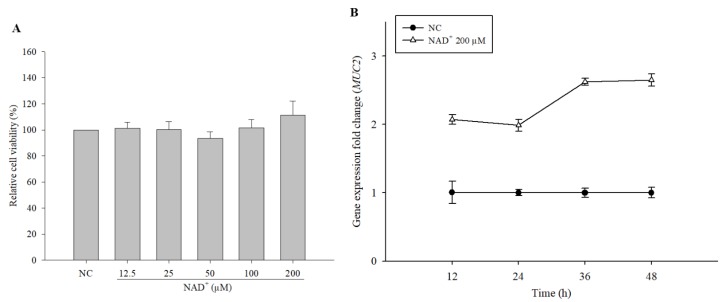
Cytotoxic effect of exogenous nicotinamide adenine dinucleotide (NAD^+^) on LS 174T goblet cells and expression level of *MUC2* by time. LS 174T cells were treated with NAD^+^ at concentrations ranging from 12.5 µM to 200 µM, and the cytotoxicity was measured by 3-(4,5-Dimethylthiazol-2-yl)-2,5-diphenyltetrazolium bromide (MTT) assay (**A**). Expression level of *MUC2* in 200 µM NAD^+^-treated cells was measured by qPCR (**B**). The data are expressed as the mean ± SD of three independent experiments performed in triplicate.

**Figure 2 biomolecules-10-00580-f002:**
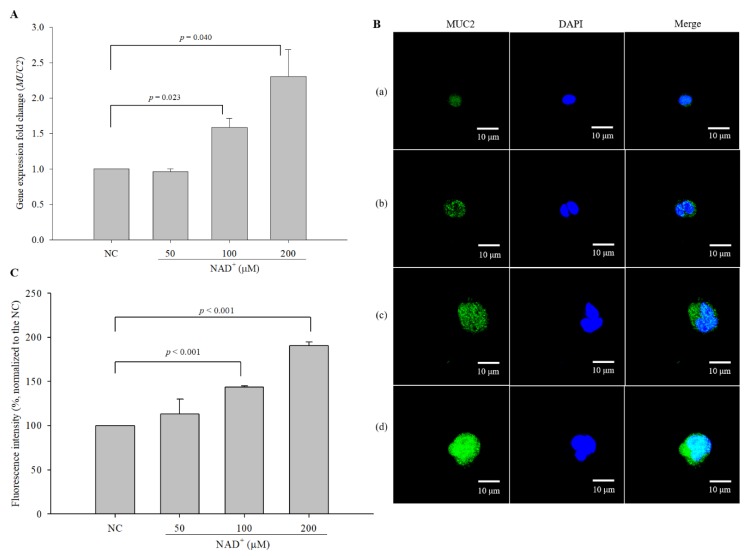
Stimulatory effect of exogenous NAD^+^ on MUC2 expression in LS 174T goblet cells. The cells were treated with NAD^+^ at concentrations ranging from 50 to 200 µM and incubated for 48 h. (**A**) The mRNA expression levels of *MUC2* were measured by qPCR. (**B**) Intracellular MUC2 protein levels were measured by immunocytochemistry (ICC) (magnification 600×), and (**C**) the intensity of fluorescence was quantified. (**a**) Negative control, (**b**) 50 µM NAD^+^, (**c**) 100 µM NAD^+^, (**d**) 200 µM NAD^+^. The data are expressed as the mean ± SD of three independent experiments performed in triplicate.

**Figure 3 biomolecules-10-00580-f003:**
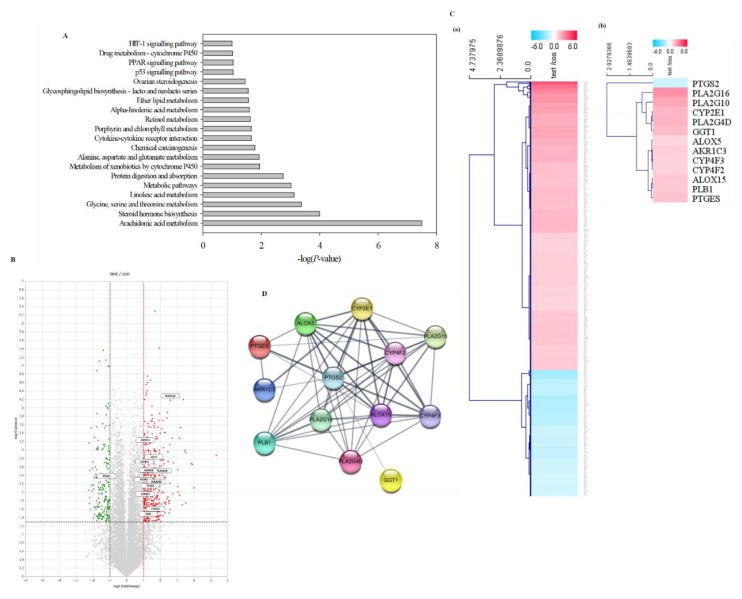
Analysis of the RNA sequencing results of cells treated with NAD^+^. RNA samples were prepared in triplicate from the negative control cells and cells treated with 200 µM NAD^+^ for 48 h, and RNA sequencing was performed using an Illumina NextSeq 500 platform. The differentially expressed genes (DEGs) were analyzed with Excel-based Differentially Expressed Gene Analysis (ExDEGA). (**A**) Functional annotation was performed with Database for Annotation, Visualization and Integrated Discovery (DAVID). (**B**) Volcano plot displaying the 648 genes with significant ≥ 2-fold and ≤ 0.5-fold (*p* < 0.05) changes in expression in 200 µM NAD^+^-treated samples compared to the negative control samples (red, significantly upregulated; green, significantly downregulated; gray, not changed significantly or changed by less than 2-fold). The boxes represent the genes involved in arachidonic acid metabolism shown in Table 2. (**C**) Heatmap based on the data of the 648 significant DEGs (Appendix A) and the genes involved in arachidonic acid metabolism (Table 2). (**D**) Interaction network of 13 arachidonic acid metabolism-associated genes (Table 2) identified by Search Tool for the Retrieval of Interacting Genes/Proteins (STRING) analysis with a confidence cutoff of 0.40 using the STRING database. In the resulting protein association network, proteins are presented as nodes connected by lines whose thickness represents the confidence level.

**Figure 4 biomolecules-10-00580-f004:**
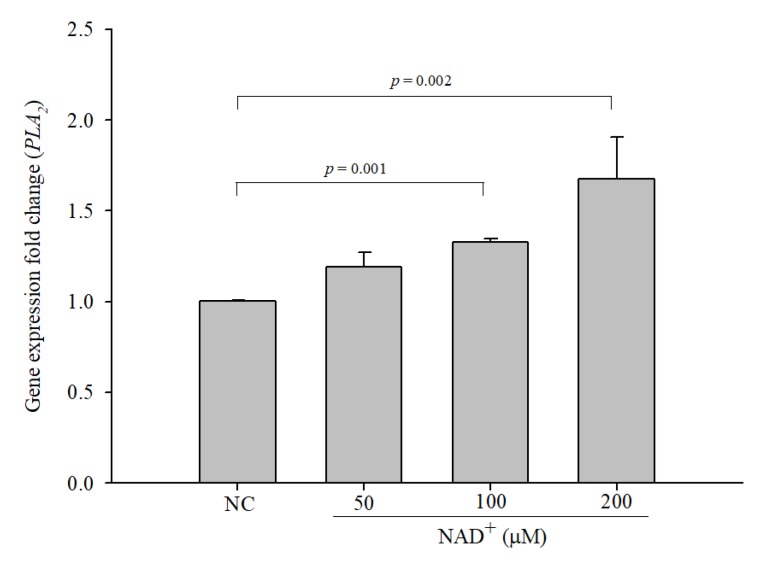
The expression levels of phospholipase A2 (PLA_2_) were increased in NAD^+^-treated cells. *PLA_2_* expression levels were measured by qPCR. The data are expressed as the mean ± SD of three independent experiments performed in triplicate.

**Figure 5 biomolecules-10-00580-f005:**
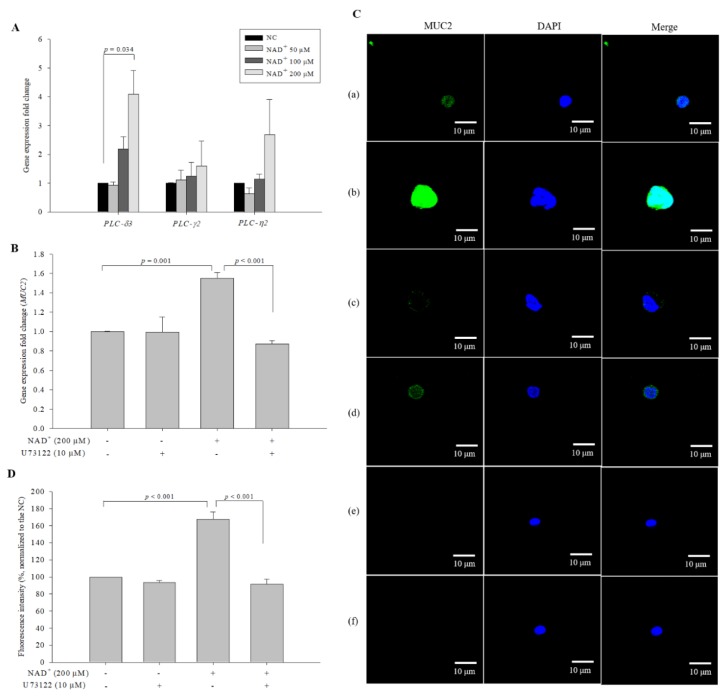
PLC-δ3 involvement in the pathway of NAD^+^-mediated stimulation of MUC2 expression. (**A**) The expression levels of *PLC* variants were detected by qPCR. U73122 was used to inhibit PLC-δ, and *MUC2* expression was observed at the gene level (**B**) and at the protein level using ICC (magnification 600×) (C). To clarify that the secondary antibodies were not affected the generation of a fluorescence signal, the preparations of the negative control and 200 µM NAD^+^-treated cells were created by omitting the primary antibody. (**a**) Negative control, (**b**) 200 µM NAD^+^, (**c**) 10 μM U73122, (**d**) 200 µM NAD^+^ and 10 μM U73122, (**e**) negative control without the primary antibody, (**f**) 200 µM NAD^+^ without the primary antibody. The fluorescence intensity was quantified (**D**). The data are expressed as the mean ± SD of three independent experiments performed in triplicate. The symbols + and − indicate chemical treatment and non-treatment, respectively.

**Figure 6 biomolecules-10-00580-f006:**
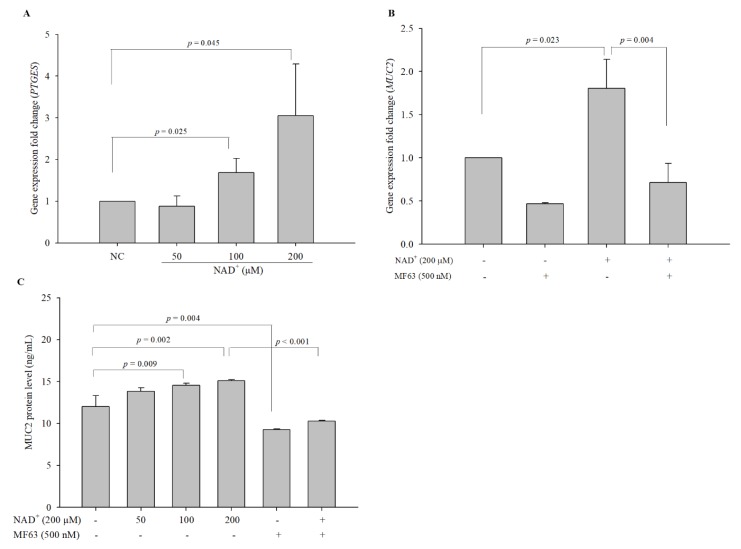
Participation of prostaglandin E synthase (PTGES) in the pathway of NAD^+^-mediated stimulation of MUC2 expression. The expression levels of *PTGES* were evaluated with qPCR (**A**). MF63 was used to inhibit PTGES. MF63 was administered to LS 174T cells alone or in combination with 200 µM NAD^+^ for 48 h and the expression levels were measured by qPCR (**B**). The MUC2 protein levels in each cell culture supernatant were assayed by ELISA (**C**). The data are expressed as the mean ± SD of three independent experiments performed in triplicate. The symbols + and − indicate chemical treatment and non-treatment, respectively.

**Figure 7 biomolecules-10-00580-f007:**
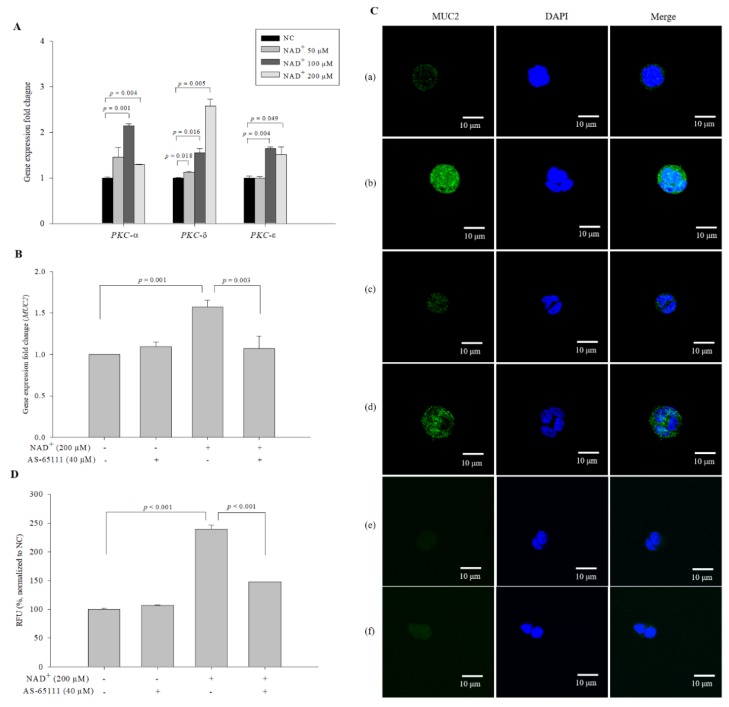
Effect of protein kinase C (PKC)-δ activation on MUC2 expression. (**A**) The expression levels of *PKC-α*, *PKC-δ*, and *PKC-ε* were investigated with qPCR. AS-65111 was used to inhibit PKC-δ. MUC2 expression was measured at the gene level (**B**) and at the protein level using ICC (magnification 600×) (**C**). To clarify that the secondary antibodies were not affected by the generation of the fluorescence signal, the preparations of the negative control and 200 µM NAD^+^-treated cells were created by omitting the primary antibody. (**a**) Negative control, (**b**) 200 µM NAD^+^, (**c**) 40 μM AS-65111, (**d**) 200 µM NAD^+^, and 40 μM AS-65111, (**e**) negative control without the primary antibody, (**f**) 200 µM NAD^+^ without the primary antibody. The fluorescence intensity was quantified (**D**). The data are expressed as the mean ± SD of three independent experiments performed in triplicate. The symbols + and − indicate chemical treatment and non-treatment, respectively.

**Figure 8 biomolecules-10-00580-f008:**
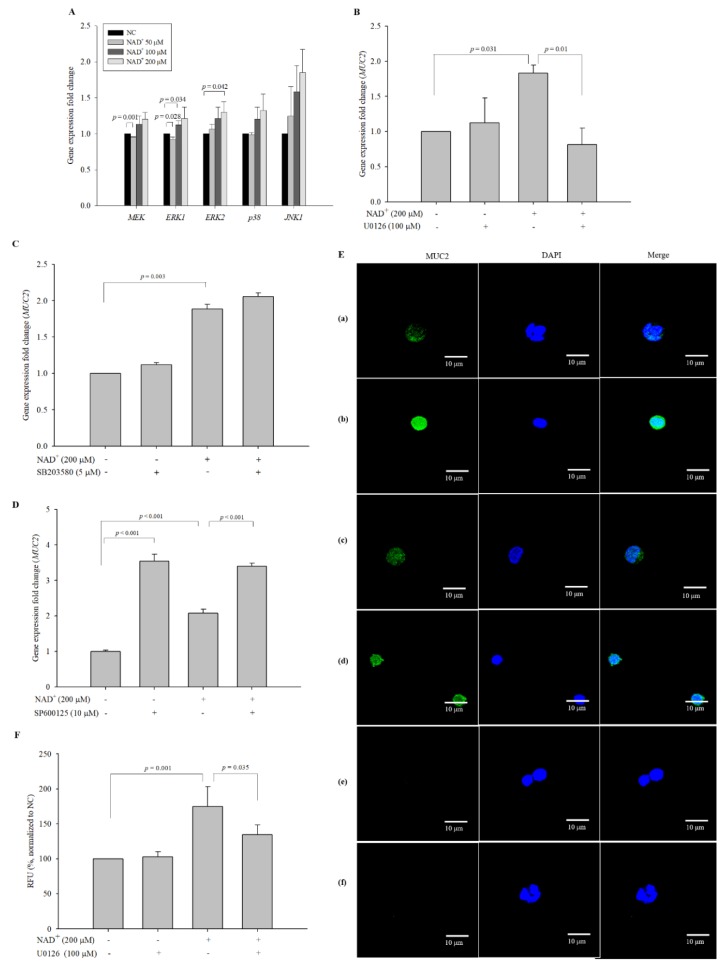
Extracellular signal-regulated kinase (ERK)1/2 is involved in *MUC2* expression in NAD^+^-treated cells. (**A**) qPCR was performed to measure the expression levels of *MEK*, *ERK1*, *ERK2*, *p38*, and *JNK1*. U0126 blocked the stimulation of *MUC2* expression caused by treatment with 200 µM NAD^+^ at the gene level (**B**), while SB203580 and SP600125 did not inhibit *MUC2* expression caused by treatment with 200 µM NAD^+^ (**C**,**D**). U0126 blocked the stimulation of MUC2 expression caused by treatment with 200 µM NAD^+^ at the protein level, as assessed using ICC (magnification 600×) (E). To clarify that the secondary antibodies were not affected the generation of fluorescence signal, the preparations of the negative control and 200 µM NAD^+^-treated cells were created by omitting the primary antibody. (**a**) Negative control, (**b**) 200 µM NAD^+^, (**c**) 100 μM U0126, (**d**) 200 µM NAD^+^, and 100 μM U0126, (**e**) negative control without the primary antibody, (**f**) 200 µM NAD^+^ without the primary antibody. The fluorescence intensity was quantified (**F**). The data are expressed as the mean ± SD of three independent experiments performed in triplicate. The symbols + and − indicate chemical treatment and non-treatment, respectively.

**Figure 9 biomolecules-10-00580-f009:**
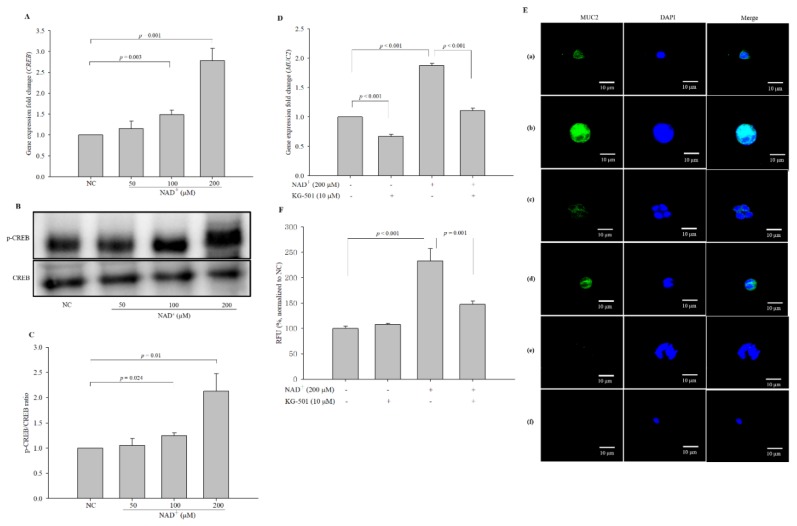
Cyclic AMP (cAMP) response element-binding protein (CREB) acts as a transcription factor to stimulate the expression of MUC2 upon exogenous NAD^+^ administration. The effect of NAD^+^ on the expression of CREB was measured by qPCR (**A**). The protein levels of CREB and phosphorylated CREB were assessed by Western blot analysis (**B**), and the p-CREB/CREB ratio was obtained (**C**). The blots are representative of three independent experiments. KG-501 was used to inhibit CREB activity, and MUC2 expression was measured at the gene level by qPCR (**D**) and the protein level by ICC (magnification 600×) (**E**). To clarify that the secondary antibodies were not affected by the generation of a fluorescence signal, the preparations of the negative control and 200 µM NAD^+^-treated cells were created by omitting the primary antibody. (**a**) Negative control, (**b**) 200 µM NAD^+^, (**c**) 10 μM KG-501, (**d**) 200 µM NAD^+^ and 10 μM KG-501, (**e**) negative control without the primary antibody, (**f**) 200 µM NAD^+^ without the primary antibody. The fluorescence intensity was quantified (**F**). The data are expressed as the mean ± standard SD of three independent experiments performed in triplicate. The symbols + and − indicate chemical treatment and non-treatment, respectively.

**Table 1 biomolecules-10-00580-t001:** Primers used for qPCR analysis.

Gene	Forward (5′ to 3′)	Reverse (5′ to 3′)
*β-actin*	GGA CTT CGA GCA AGA GAT GG	AGC ACT GTG TTG GCG TAC AG
*MUC2*	ACC CGC ACT ATG TCA CCT TC	GGA CAG GAC ACC TTG TCG TT
*PLC-δ3*	TCT CTT CCT CCC ACA ACA CC	TCC AGG GAT AGG ATG ACA GG
*PLC-γ2*	CGT CTA CCC AAA GGG ACA AA	GAC TGT CAG CGT CAT CAG GA
*PLC-η2*	GCC ACC CAC GAC ATA GAG AT	ACG GTA GGA GGA GGG GTA GA
*PKC-α*	AGC CCA AAG TGT GTG GCA AA	AGG TGT TTG TTC TCG CTG GT
*PKC-δ*	CCC TTC TGT GCC GTG AAG AT	GCC CGC ATT AGC ACA ATC TG
*PKC-ε*	GAA CCC GGC GAG GAA ATA CA	AGG GCA GGA ATG AAG AAC CG
*MEK*	GCT TGG GGC TAT TTG TGT GT	TCT CAC AAG GCT CCC TCC TA
*ERK1*	TCA GAC TCC AAA GCC CTT GA	CGT GCT GTC TCC TGG AAG AT
*ERK2*	TCC AAC AGG CCC ATC TTT CC	CCA GAG CTT TGG AGT CAG CA
*PTGES*	CAT GTG AGT CCC TGT GAT GG	GAC TGC AGC AAA GAC ATC CA
*PLA2*	TGG CTC TGT GTG ATC AGG AG	GAG CCA GAA AGA CCA GCA AC
*CREB*	CTG CCT CTG GAG ACG TAC AA	CAA GCA CTG CCA CTC TGT TT
*JNK-1*	GCT TGG AAC ACC ATG TCC TG	GTA CGG GTG TTG GAG AGC TT
*p38*	GGG GCA GAT CTG AAC AAC AT	CAG GAG CCC TGT ACC ACC TA

**Table 2 biomolecules-10-00580-t002:** Genes involved in arachidonic acid metabolism network by STRING analysis.

Gene	Gene Name	Expression Fold Change ^a^	*P*-Value
*PLA2G10*	Phospholipase A2 group X	4.323	0.004
*GGT1*	Gamma-glutamyltransferase 1	3.061	0.002
*PLB1*	Phospholipase B1	2.437	0.043
*AKR1C3*	Aldo-keto reductase family 1 member C2	2.158	0.001
*PTGES*	Prostaglandin E synthase	2.662	0.009
*CYP2E1*	Cytochrome P450 family 2 subfamily E member 1	3.295	0.032
*ALOX5*	Arachidonate 5-lipoxygenase	2.012	0.006
*ALOX15*	Arachidonate 15-lipoxygenase	2.469	0.004
*PLA2G4D*	Phospholipase A2 group IVD	3.406	0.007
*PLA2G16*	Phospholipase A2 group XVI	6.022	< 0.001
*CYP4F3*	Cytochrome P450 family 4 subfamily F member 3	2.139	0.014
*CYP4F2*	Cytochrome P450 family 4 subfamily F member 2	2.087	0.002
*PTGS2 (COX-2)*	Prostaglandin-endoperoxide synthase 2 (cyclooxygenase-2)	0.433	0.005

^a^ Expression fold change: fold change in the expression a gene differentially expressed in 200 μM NAD^+^-treated cells versus the negative control cells.

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
