# Peer review of "Exogenous NAD+ Stimulates MUC2 Expression in LS 174T Goblet Cells via the PLC-Delta/PTGES/PKC-Delta/ERK/CREB Signaling Pathway"

_biomolecules, 2020, doi:10.3390/biom10040580_

Round 1

Reviewer 1 Report

The manuscript submitted by Ma et and colleagues describes the role of extracellular NAD+ in the in vitro upregulation of MUC2 in a human goblet cell line (LS 174T). The paper is well written and structured and the authors move hand over hand along the NAD+ triggered signaling pathway that leads via a yet unknown receptor along PLC-delta, PTGES, PKC-4 delta and ERK to the phosphorylation of CREB with is a major regulator of MUC2 expression and mucus production. The authors analyses were performed on the mRNA and protein expression level, for the latter they used quantified fluorescence microscopy. Though the most interesting questions such as “What is the receptor for NAD on goblet cells” or “Where does the NAD+ come from” still need to be addressed in the future, the authors reveal one important part of the puzzle in this manuscript, which is pinpointing down the signaling pathway that NAD+ triggers in the goblet cell line. I recommend this manuscript for publication after careful editing for typos etc such as: page 6 line 222 “performed in three independent experiments” or page 8 line 260 “PLA2 converts membrane phospholipids from into arachidonic acid.”

Author Response

We deeply appreciate your kind and reasonable comments.

According to your suggestion, we revised the manuscript carefully and marked the corrections in red.

I recommend this manuscript for publication after careful editing for typos etc such as: page 6 line 222 “performed in three independent experiments” or page 8 line 260 “PLA2 converts membrane phospholipids from into arachidonic acid.”

â–º According to your comment, we corrected the sentences. We added “in” (line 223) and deleted “from” (line 261).

Reviewer 2 Report

This manuscript tries to elucidate the mechanisms by which NAD+ stimulates MUC2 expression in LS 174T goblet cells. Overall, methods employed are described detailed enough and results are convincing. The stimulation of MUC2 expression by NAD+ is at most two-fold. Whether NAD+ is a biologically significant signaling molecule in this context should be discussed. The involvement of CREB in the stimulation of MUC2 gene expression is presented. Also, PGE2 involvement is suggested. However, the cAMP pathway is not considered. This could be one weakness of the manuscript. The text in the result section is mostly a redundant description of figures. This section should have more interpretation of the results.

I have the following specific comments/questions for the authors to consider.

L94. Culture vessel size and medium volume should be listed.

L100-L107. Were cells growing (increase in number) during the 48 h incubation? Depending on the initial cell number this can be proliferation, not viability assay.

L128. Which software was used for image analysis?

L145. DEG should be defined at its first appearance.

L147-L152. Text is redundant needs editing.

Many figures have “p<0.000”. This does not make sense and needs to be edited.

L214. 12.5 should be 50.

Figure 3D. Is there any meaning of difference in color?

Table 2 shows a p-value of 0.000, which does not make sense need editing.

L260. Typo in the first sentence (“from into”).

L261. There are three PLA2 genes listed in Table 2. Which one was quantified? Table 1 shows only one set of primers for PLA2.

Many numbers in the text are shown to third decimal place (ex 1.191, 1.329, and 1.677 in L264). What the rationale for this? Are these numbers accurate to the third decimal place?

L271. This sentence needs a reference.

L315-L316. A secondary antibody with a different fluorescent dye (red) can be used in ICC.

Figure 6B. MF63appeared to significantly decrease MUC2 expression compared to control. This should be discussed.

Figure 6C. Typo on rightmost bar legend. + should be 200.

L331. This needs a reference. PGE2 can activate PKA by increasing cAMP, too.

Figure 8D. SP600125 significantly increased MUC2 expression. This should be mentioned and discussed.

Figure 9D. KG-501 significantly decrease MUC2 expression compared to control. This should be discussed.

L429-L443. There is no need for an increase of P2Y receptor gene expression for NAD+ to activate PLC through the receptors. Table S1 shows the expression of P2Y11, which can be activated by NAD+. Discussion of GPR87 and PGR34 is far-fetched.

L473-L475. The text includes contradictory statements and needs editing.

Reviewer 3 Report

The paper on “Exogenous NAD+ Stimulates MUC2 Expression in LS 174T Goblet Cells via the PLC-delta/PTGES/PKC-delta/ERK/CREB Signaling Pathway” reported on interesting data regarding the defence system of mucus layer intestine.

The authors demonstrated in their manuscript that exogenous NAD+ induces the activation of PLC-δ/PTGES/PKC-δ/ERK1/2 signaling to phosphorylate the transcription factor CREB, which enhances MUC2 expression in intestinal goblet cells.

From their findings, the authors conclude that NAD+ has the potential to improve intestinal mucosal barrier function by stimulating the expression of MUC2.

.

The study was well designed. Also, the paper was nicely written, and the authors used a good range of modern technologies to corroborate their hypothesis.

Author Response

We deeply appreciate your kind and reasonable comments.